# Prospective Investigation of Surgical Outcomes after Anterior Decompression with Fusion and Laminoplasty for the Cervical Ossification of the Posterior Longitudinal Ligament: A Propensity Score Matching Analysis

**DOI:** 10.3390/jcm11237012

**Published:** 2022-11-27

**Authors:** Toshitaka Yoshii, Shingo Morishita, Satoru Egawa, Kenichiro Sakai, Kazuo Kusano, Shunji Tsutsui, Takashi Hirai, Yu Matsukura, Kanichiro Wada, Keiichi Katsumi, Masao Koda, Atsushi Kimura, Takeo Furuya, Satoshi Maki, Narihito Nagoshi, Norihiro Nishida, Yukitaka Nagamoto, Yasushi Oshima, Kei Ando, Hiroaki Nakashima, Masahiko Takahata, Kanji Mori, Hideaki Nakajima, Kazuma Murata, Masayuki Miyagi, Takashi Kaito, Kei Yamada, Tomohiro Banno, Satoshi Kato, Tetsuro Ohba, Satoshi Inami, Shunsuke Fujibayashi, Hiroyuki Katoh, Haruo Kanno, Hiroshi Taneichi, Shiro Imagama, Yoshiharu Kawaguchi, Katsushi Takeshita, Morio Matsumoto, Masashi Yamazaki, Atsushi Okawa

**Affiliations:** 1Department of Orthopedic Surgery, Graduate School of Medicine, Tokyo Medical and Dental University, Tokyo 113-8519, Japan; 2Department of Orthopedic Surgery, Saiseikai Kawaguchi General Hospital, Saitama 332-8558, Japan; 3Department of Orthopedic Surgery, Kudanzaka Hospital, Chiyadaku 102-0074, Japan; 4Department of Orthopaedic Surgery, Faculty of Medicine, Wakayama Medical University, Wakayama 641-8510, Japan; 5Department of Orthopedic Surgery, Hirosaki University Graduate School of Medicine, Aomori 036-8562, Japan; 6Department of Orthopedic Surgery, Niigata University Medical and Dental General Hospital, Niigata 951-8520, Japan; 7Department of Orthopedic Surgery, Faculty of Medicine, University of Tsukuba, 1-1-1 Tennodai, Tsukuba, Ibaraki 305-8575, Japan; 8Department of Orthopedics, Faculty of medicine, Jichi Medical University, Tochigi 329-0498, Japan; 9Department of Orthopedic Surgery, Chiba University Graduate School of Medicine, Chiba 260-0856, Japan; 10Department of Orthopaedic Surgery, School of Medicine, Faculty of Medicine, Keio University, Tokyo 160-8582, Japan; 11Department of Orthopedic Surgery, Yamaguchi University School of Medicine, Yamaguchi 755-8505, Japan; 12Department of Orthopedic Surgery, Osaka Rosai Hospital, Osaka 591-8025, Japan; 13Department of Orthopaedic Surgery, Faculty of Medicine, The University of Tokyo, Tokyo 113-0033, Japan; 14Department of Orthopedic Surgery, Nagoya University Graduate School of Medicine, Aichi 466-8550, Japan; 15Department of Orthopaedic Surgery, Faculty of Medicine and Graduate School of Medicine, Hokkaido University, Sapporo 060-8638, Japan; 16Department of Orthopaedic Surgery, Faculty of Medicine, Shiga University of Medical Science, Shiga 520-2192, Japan; 17Department of Orthopaedics and Rehabilitation Medicine, Faculty of Medical Sciences, University of Fukui, Fukui 910-1193, Japan; 18Department of Orthopedic Surgery, Faculty of Medicine, Tokyo Medical University, Tokyo 160-0023, Japan; 19Department of Orthopedic Surgery, Faculty of Medicine, Kitasato University Hospital, Kanagawa 252-0375, Japan; 20Department of Orthopedic Surgery, Graduate School of Medicine, Osaka University, Osaka 565-0871, Japan; 21Department of Orthopaedic Surgery, Kurume University School of Medicine, Fukuoka 830-0011, Japan; 22Department of Orthopedic Surgery, Hamamatsu University School of Medicine, Shizuoka 431-3125, Japan; 23Department of Orthopaedic Surgery, Graduate School of Medical Sciences, Kanazawa University, Kanazawa 920-8641, Japan; 24Department of Orthopedic Surgery, Faculty of Medicine, University of Yamanashi, Yamanashi 409-3898, Japan; 25Department of Orthopaedic Surgery, Dokkyo Medical University School of Medicine, Tochigi 321-0293, Japan; 26Department of Orthopaedic Surgery, Graduate School of Medicine, Kyoto University, Kyoto 606-8507, Japan; 27Department of Orthopedic Surgery, Surgical Science, Tokai University School of Medicine, Kanagawa 259-1193, Japan; 28Department of Orthopaedic Surgery, Tohoku University School of Medicine, Miyagi 980-8574, Japan; 29Department of Orthopedic Surgery, Faculty of Medicine, University of Toyama, Toyama 930-0194, Japan

**Keywords:** ossification of the posterior longitudinal ligament, anterior decompression and fusion, laminoplasty, propensity score matching, neurological recovery, minimum clinically important difference

## Abstract

The ideal surgical strategy for cervical ossification of the posterior longitudinal ligament (OPLL) remains controversial due to the lack of high-quality evidence. Herein, we prospectively investigated the surgical outcomes of anterior cervical decompression with fusion (ADF) and laminoplasty (LAMP) with cervical OPLL. Three hundred patients were included in this study (ADF: *n* = 89; LAMP: *n* = 211 patients), and propensity score matching yielded 67 pairs of patients with ADF and LAMP, in which clinical outcomes were compared. Crude analysis revealed that the ADF group showed greater neurological recovery in cervical Japanese Orthopedic Association scores at two years, compared with that in the LAMP group (53.1% vs. 44.3%, *p* = 0.037). The ratio of minimum clinically important difference (MCID) success was significantly greater in the ADF group (59.6% vs. 43.6%, *p* = 0.016). Multivariate analysis showed that the factors affecting MCID success were age, body mass index, duration of symptoms, and choice of ADF. In the 1:1 matched analysis, neurological improvement was more favorable in the ADF group (57.2%) compared to the LAMP group (46.8%) at two years (*p* = 0.049). However, perioperative complications, such as dysphagia and graft-related complications, were more common in the ADF group.

## 1. Introduction

The Ossification of the posterior longitudinal ligament (OPLL) is a common degenerative spine disease that causes neurological dysfunction, and it is especially prevalent in many Asian countries [1]. In the early stages of the disease, most patients may not exhibit any neurological symptoms. However, with the development of OPLL in the spinal canal, the spinal cord and nerve roots are compressed from the anterior direction, inducing symptomatic myelopathy. Minimally symptomatic patients can be treated conservatively; however, patients with progressive myelopathy require immediate surgical intervention [2].

The optimal surgical procedure for treating patients with cervical OPLL remains controversial. Anterior decompression and fusion (ADF) can provide direct decompression to the spinal cord and stabilize the involved segments [3,4]. However, this procedure is complex and technically demanding. Posterior decompression techniques, such as laminoplasty (LAMP), are relatively less technically demanding, but an indirect decompression may often be insufficient, resulting in poor neurological recovery [2,5]. Although many previous studies have compared the clinical outcomes of patients after ADF and LAMP, most of them were retrospective studies with small sample sizes or meta-analysis of these studies [2,5,6,7]. Thus, there is insufficient evidence regarding the optimal surgical approach for the treatment of cervical OPLL.

In the present study, we conducted a nationwide study on ADF and LAMP for patients with cervical OPLL, investigating the clinical outcomes of each surgical method. Neurological recovery, pain scores, and perioperative complications after ADF or LAMP were evaluated. Furthermore, the surgical outcomes in propensity score matched patients were also compared to minimize bias between the groups. To our knowledge, this is the largest prospective cohort study to compare ADF and LAMP for patients with cervical OPLL.

## 2. Materials and Methods

### 2.1. Participants

This multi-institutional study on cervical OPLL surgeries was performed using the data from the Japanese Multicenter Research Organization for Ossification of the Spinal Ligament (JOSL) with the assistance of the Japanese Ministry of Health, Labor, and Welfare. This study was approved by all 28 institutions belonging to the JOSL. Three hundred patients with neurological disturbance caused by cervical OPLL who underwent ADF (*n* = 89 patients) or LAMP (*n* = 211 patients) from 2014 to 2017 were investigated and were prospectively followed-up for two years.

In this study, patients were eligible for inclusion if they (1) were aged 20 years or older, (2) had evidence of OPLL based on computed tomography and spinal cord compression on magnetic resonance imaging (MRI), and (3) underwent ADF or LAMP. All patients underwent cervical spine surgery, and the approach was selected at the discretion of surgeons and patients. Patients were excluded if they had a history of cervical surgery, anterior-posterior combined surgery, posterior instrumented fusion, and neurological disturbances due to disk herniation, infection, trauma, or spondylosis. We confirmed that all methods were carried out in accordance with relevant guidelines and regulations. Informed consent was obtained from all subjects. 

### 2.2. Evaluation

Data were collected from each patient, including demographic information, medical history, and radiological findings. Clinical outcomes were assessed before and after surgery using the cervical Japanese Orthopedic Association (JOA) [8] and visual analog scale (VAS) scores. The JOA recovery rate was calculated as follows: (postoperative JOA − preoperative JOA)/(17 − preoperative JOA) × 100 [9]. The MCID in the JOA recovery rate utilized was 52.8% [10,11]. 

### 2.3. Radiological Assessments

The types of OPLL were classified as localized, segmental, continuous, and mixed [12]. The Cobb angle between C2 and C7, thickness, occupying ratio of OPLL, and K-line were evaluated by preoperative plain lateral radiography taken in a neutral position. K-line is a line that connects the midpoints of the spinal canal between C2 and C7 on a plain lateral radiograph, and K-line (−) is defined when the peak of the ossified lesions exceeds that line [13]. Graft-related complications included graft subsidence (≥2 mm), graft dislodgement (≥2 mm) as well as hardware failure such as plate dislodgement and screw back-out.

### 2.4. Statistics

Continuous variables were presented as mean ± standard deviation, while categorical variables were presented as percentages. Unpaired Student’s t tests were used to compare continuous variables, chi-square tests or Fisher’s exact tests while were used for categorical variables. Logistic regression analysis was performed to evaluate the factors affecting the success in achieving the MCID of JOA scores. A *p* value < 0.05 was considered statistically significant. All statistical analyses were performed using SPSS v. 26.0 (SPSS Inc., Chicago, IL, USA).

The propensity scores for the surgical procedures were calculated based on the patients’ age, sex, BMI, preoperative neurological status, duration of symptoms, comorbidities, and K-line. The C-statistic suggested that the fitting was 0.67. Patients that underwent ADF or LAMP were matched based on the propensity scores on a condition that the caliper was lower than 0.2.

## 3. Results

### 3.1. Demographics

Table 1 shows the demographic parameters of patients included in this study. There were no significant differences in sex, BMI, duration of symptoms, smoking history, comorbidities, or anticoagulant drug use between the ADF and LAMP groups. However, patients were younger in the ADF group than in the LAMP group (*p* = 0.001), and more patients with comorbidities were found in the LAMP group (*p* = 0.010). Furthermore, the JOA scores revealed no significant differences in the preoperative neurological status between the two groups. There was no significant difference in the radiological classification of OPLL or frequency of increased signal intensities on MRI. Patients with large OPLL and K-line (−) were more frequently found in the ADF group (*p* < 0.001).

### 3.2. Perioperative Complications

Perioperative complications are summarized in Table 2. Approach-related complications, such as upper airway obstruction and dysphagia, were more common in the ADF group than in the LAMP group (3.4% vs. 0%, *p* = 0.025; 7.9% vs. 0%, *p* < 0.001, respectively). Local complications, such as dural tear and cerebrospinal fluid (CSF) leak, were also more common in the ADF group (10.1% vs. 2.8%, *p* = 0.009; 4.5% vs. 1.9%, *p* = 0.012, respectively). In addition, graft-related complications were more frequently observed in the ADF group than in the LAMP group (9.0% vs. 0%, *p* < 0.001). In contrast, our findings showed that the rate of surgical site infection (SSI) was significantly higher in the LAMP group than in the ADF group (1.1% vs. 2.4%, *p* = 0.022). There were no significant differences in the occurrence of unilateral segmental motor palsy (C5 palsy) between the two groups. Reoperation was more frequent in the ADF group. There were 8 patients who received reoperation in the ADF group. Additional decompression was required for C5 palsy in 1 patient and for adjacent segmental disease in 1 patient. Hematoma removal was performed in 1 patient. Five patients required revision surgeries due to graft-related complications: anterior revision was performed for 2 patients and anterior revision followed by posterior fusion was required for 3 patients.

### 3.3. Surgical Outcomes

Surgical outcomes are shown in Table 3. The neurological improvement rate assessed by the cervical JOA score was 52.6% ± 30.8% in the ADF group and 45.9% ± 30.6% in the LAMP group one year after the operation (*p* = 0.088). The recovery rate in the ADF group was 53.1% ± 31.1% after two years, which was significantly higher than that in the LAMP group (44.3% ± 33.7%; *p* = 0.037). Furthermore, the ratio of minimum clinically important difference (MCID) success in JOA scores was significantly greater in the ADF group (59.6% vs. 43.6%, *p* = 0.016). There were no significant differences between the ADF and LAMP groups in terms of pain/numbness VAS scores in the neck, chest, upper extremities, or lower extremities.

### 3.4. Factors Affecting the MCID Success

Table 4 demonstrates the factors that influence the success of MCID. One hundred 45 patients (48.3%) achieved MCID success based on the JOA scores two years after the operation. The results showed that age (OR: 0.951, 95% CI: 0.927–0.976, *p* < 0.001), BMI (OR: 0.910, 95% CI: 0.851–0.973, *p* = 0.006), duration of symptoms (OR: 0.995, 95% CI: 0.991–1.000, *p* = 0.034), and choice of ADF (OR: 1.119, 95% CI: 1.038–3.149, *p* = 0.037) were significant factors related to the postoperative achieving MCID. Other factors, such as comorbidities, pre-JOA score, and K-line, were not significantly associated with the MCID success.

### 3.5. Comparison of the Matched Patients

Table 5 shows the comparison of 1:1 matched patients in the ADF (*n* = 67 patients) and LAMP (*n* = 67 patients) groups. There were no differences in patients’ demographics between the two groups, while neurological improvement was more favorable in the ADF group compared to the LAMP group one year (54.6 ± 30.0 vs. 43.7 ± 30.2, *p* = 0.042) and two years after the surgery (57.2 ± 30.5 vs. 46.8 ± 28.2, *p* = 0.049).

Perioperative complications, such as dysphagia and graft-related complications, were more common in the ADF group (9.0% vs. 0%, *p* = 0.025; 10.4% vs. 0%, *p* = 0.001). The incidence of unilateral segmental motor palsy was slightly higher in the ADF group (6.0% vs. 1.5%, *p* = 0.18), whereas the incidence of SSI was higher in the LAMP group (1.5% vs. 6.0%, *p* = 0.091). Assessment of pain intensity using VAS scores also showed no significant differences in the neck, chest, upper extremities, or lower extremities of patients in both groups.

## 4. Discussion

Both ADF and LAMP are widely applied for the treatment of cervical OPLL [3,14]. The present prospective, multicenter study compared the surgical results and clinical outcomes of ADF and LAMP for cervical OPLL. Demographics showed that the patients who received ADF were younger and had more comorbidities. A previous study reported that ADF was associated with a higher incidence of perioperative complications compared to LAMP [15]. Surgeons are more likely to choose LAMP for high-risk patients, such as patients of advanced age and/or comorbidities. Regarding the radiological characteristics, the ratio of K-line (−) OPLL in the ADF group was significantly higher than that in the LAMP group. It was reported that K-line (−) patients could experience insufficient decompression and poor neurological recovery when treated by LAMP [13]. Therefore, the percentage of K-line (−) patients in the ADF group was almost twofold greater than that in the LAMP group. Our findings clearly demonstrate a trend in surgical procedure of choice for cervical OPLL in this nationwide study in Japan.

The incidence of perioperative complications was higher in the ADF group than that in the LAMP group, and this tendency was similar even after the analysis of matched patients. Approach-related complications, such as upper airway obstruction and dysphagia, were more common in the ADF group. In the anterior approach, the esophagus was retracted during the surgery. Dysphagia can occur due to esophageal irritation, laryngeal nerve impairment, or retropharyngeal wall edema, which can also cause airway obstruction. Previous studies have also reported that the incidence of postoperative dysphagia was markedly higher in the anterior approach than that in the posterior one [15,16]. In addition, as airway obstruction can cause devastating fatal events, extensive care should be taken on the retropharyngeal swelling after an anterior approach for cervical OPLL.

Local complications, including dural tear and CSF leak, were more frequently observed in the ADF group than in the LAMP group. In patients with OPLL, the dura mater becomes ossified and adheres to the posterior longitudinal ligament, which increases the risk for durotomy during anterior decompression [17]. Thus, the anterior approach is high-risk for CSF leak: 6.7–31.8% [9,18,19,20]. In contrast, the floating method or anterior sliding osteotomy may be an effective procedure to reduce the risk of CSF leak and related complications [21,22]. Although most perioperative complications were more common in the ADF group, SSI more frequently occurred in the LAMP group. Previous studies have also shown that the SSI rate after posterior cervical spine surgery was higher than that observed following the anterior approach [15,23]. Compared to the posterior approach, the anterior approach has the advantages of minimal soft tissue and muscle dissection, resulting in a lower incidence of SSI after ADF. Patients with OPLL commonly suffer from comorbidities, such as obesity and DM [10], which are both associated with a higher risk for SSI. Therefore, the anterior procedure may be preferable for patients who are susceptible to SSI.

With respect to the neurological improvement after surgery, our crude analysis revealed that the ADF group showed a superior recovery rate of JOA scores compared to that in the LAMP group. The rate of patients who achieved MCID in neurological improvement was 59.6% in the ADF group, which was significantly higher than that in the LAMP group (43.6%). Moreover, the multivariate analysis showed that ADF was one of the independent factors affecting MCID success, indicating that ADF is a more effective approach. The patients undergoing ADF and LAMP were further matched using the PS 1:1 matching method. Classical 1:1 caliper matching without replacement is simple and can produce estimates with lower bias and better nominal coverage than matching with replacement when 1:1 PS matching was considered with enough number of samples. As a result, all factors correlated to neurological outcomes in the multivariate analysis (i.e., age, BMI, and duration of symptoms) were matched well between the two groups. In this PS matching analysis, neurological improvement was more favorable in the ADF group compared to the LAMP group, confirming the superiority of ADF for a favorable neurological recovery. The difference looks more marked in the early postoperative period (at six months after surgery: *p* = 0.017). A previous prospective study also indicated that ADF showed a greater neurological improvement than LAMP for cervical OPLL [5]. Since the spinal cord compression is always located at the anterior side of patients with OPLL, anterior direct decompression is a reasonable procedure, which could facilitate early and effective improvement of neurological symptoms. 

The present study has several limitations. First, as this is a multicenter study, surgical indications and postoperative protocols may have varied among study hospitals, subsequently influencing surgical outcomes. Second, our study was not randomized, thus selection of surgical procedures could be biased. Although multivariate analysis and PS matching analysis were performed to diminish biases, there may be residual differences for other measured and unmeasured baseline covariates. Additionally, it was technically difficult to put many covariates which closely correlated with each other for propensity score calculating. Third, we evaluated C2-7 alignment but did not assess local alignment at the OPLL, which may influence the surgical results.

Despite these limitations, the present study is the first to compare the surgical outcomes of ADF and LAMP for patients with OPLL using PS matching analysis of prospectively collected data. Our findings will help surgeons select an operative procedure and make appropriate informed decisions for patients with OPLL.

## 5. Conclusions

Our analysis showed that ADF and LAMP could both substantially improve the neurological symptoms in patients with cervical OPLL. ADF demonstrated a greater improvement in neurological score compared to LAMP. Patients undergoing ADF showed a significantly higher rate of MCID success. In contrast, perioperative complications, such as dysphagia and graft-related complications, were more common in the ADF group than in the LAMP group.

## Figures and Tables

**Table 1 jcm-11-07012-t001:** Demographics.

		ADF (*n* = 89)	LAMP (*n* = 211)	*p* Value
Age (year)		60.1 ± 10.9	65.1 ± 11.7	0.001 *
Sex	(male ratio: %)	68.5%	74.9%	0.26
Body mass index		25.8 ± 4.3	25.2 ± 3.9	0.25
Smoking history		41.6%	37.0%	0.45
Duration of symptoms		46.3 ± 56.7	43.9 ± 70.5	0.78
Pre JOA score		10.8 ± 2.7	10.9 ± 2.9	0.86
Comorbidities		67.4%	81.0%	0.010 *
	diabetes mellitus	23.6%	29.9%	0.23
	hypertension	31.5%	42.7%	0.11
	malignancy	4.5%	5.7%	0.79
	cerebrovascular disease	2.2%	7.6%	0.050
	myocardial infarction (%)	4.5%	1.9%	0.26
	collagen disease	1.1%	0.5%	0.47
Drug	anticoagulant use	11.2%	15.2%	0.47
Neurological evaluation	Pre JOA score	10.9 ± 2.6	11.0 ± 2.7	0.86
Radiological classification	Seg/Cont/Mix/Loc	38.2/12.4/41.6/7.9%	45.0/11.4/35.5/8.1%	0.72
Alignment	Lordosis (≥0°)	80.9%	88.6%	0.097
K-line	K-line (−)	46.1%	22.3%	<0.001 *
Occupying ratio	≤20/between 20 and 40/between 40 and 60/between 60 and 75/>75 (%)	2.2/28.1/47.2/16.9/5.6%	8.1/53.1/31.3/6.2/1.4%	<0.001 *
MRI	T2 high intensity (+)	86.5%	85.3%	0.79

ADF: anterior decompression with fusion; LAMP: laminoplasty; JOA: Japanese Orthopedic Association; Seg: segmental type/Cont: continuous type/Mix: mixed type/Loc: localized type; MRI: magnetic resonance imaging; *: *p* < 0.05.

**Table 2 jcm-11-07012-t002:** Perioperative complications in ADF and LAMP (crude analysis).

	ADF (*n* = 89)	LAMP (*n* = 211)	*p* Value
Surgical site infection	1.1%	2.4%	0.022 *
Dural tear	10.1%	2.8%	0.009 *
CSF leakage	4.5%	1.9%	0.012 *
Graft-related complications	9.0%	0%	<0.001 *
Upper airway obstruction	3.4%	0%	0.025 *
Dysphagia	7.9%	0%	<0.001 *
C5 palsy	5.6%	5.7%	0.53
Reoperation	9.0%	1.9%	0.008 *

ADF: anterior decompression with fusion; LAMP: laminoplasty; CSF: Cerebrospinal fluid; *: *p* < 0.05.

**Table 3 jcm-11-07012-t003:** Postoperative neurological evaluation in ADF and LAMP (crude analysis).

		ADF (*n* = 89)	LAMP (*n* = 211)	*p* Value
Neurological evaluation	JOA score (6 m)	14.0 ± 2.3	13.3 ± 2.6	0.061
	JOA RR (6 m)	50.6 ± 36.4	36.4 ± 50.8	0.037 *
	JOA score (1 y)	14.1 ± 2.3	13.8 ± 2.3	0.29
	JOA RR (1 y)	52.6 ± 30.8	45.9 ± 30.6	0.088
	JOA score (2 y)	14.1± 2.4	13.7 ± 2.4	0.16
	JOA RR (2 y)	53.1 ± 31.1	44.3 ± 33.7	0.037 *
	MCID success ratio (%)	59.6%	43.6%	0.016 *
VAS (2 y)	pain or stiffness in the neck or shoulder	35.9 ± 30.2	31.6 ± 28.1	0.43
	tightness in the chest	6.8 ± 18.0	7.9 ± 15.2	0.72
	pain or numbness in the arms or hands	41.4 ± 33.2	36.4 ± 27.3	0.39
	pain or numbness from chest to toe	31.0 ± 31.9	33.3 ± 32.5	0.71

ADF: anterior decompression with fusion; LAMP: laminoplasty; JOA: Japanese Orthopedic Association; RR: recovery rate, MCID: minimum clinically important difference; VAS: visual analogue scale; *: *p* < 0.05.

**Table 4 jcm-11-07012-t004:** Factors affecting MCID Success.

	OR	95% CI	*p* Value
Age	0.951	0.927–0.976	<0.001 *
Body mass index	0.910	0.851–0.973	0.006 *
Duration of symptoms	0.995	0.991–1.000	0.034 *
Choice of ADF	1.808	1.038–3.149	0.037 *
Comorbidities (+)	1.119	0.618–2.028	0.71
Pre JOA score	0.932	0.848–1.024	0.13
K-line (−)	0.782	0.404–1.237	0.22

MCID: minimal clinically important difference; ADF: anterior decompression with fusion; LAMP: laminoplasty; JOA: Japanese Orthopedic Association; *: *p* < 0.05.

**Table 5 jcm-11-07012-t005:** Comparison of matched patients of ADF and LAMP groups.

		ADF (*n* = 67)	LAMP (*n* = 67)	*p* Value
Age (year)		60.1 ± 10.6	60.2 ± 11.4	0.93
Sex	(male ratio)	68.5%	74.9%	0.26
Body mass index		25.8 ± 4.1	25.8 ± 4.2	0.94
Duration of symptoms		47.1 ± 55.9	49.2 ± 84.8	0.86
Comorbidities		65.7%	79.1%	0.12
Pre JOA score		10.9 ± 2.6	10.5 ± 2.9	0.41
K-line	K-line (−)	41.8%	35.8%	0.48
Neurological evaluation	JOA score (6 m)	14.1 ± 2.3	13.2 ± 2.5	0.032 *
	JOA RR (6 m)	52.0 ± 30.4	39.2 ± 30.3	0.017 *
	JOA score (1 y)	14.3 ± 2.1	13.4 ± 2.4	0.029 *
	JOA RR (1 y)	54.6 ± 30.0	43.7 ± 30.2	0.042 *
	JOA score (2 y)	14.4 ± 2.3	13.6 ± 2.5	0.042 *
	JOA RR (2 y)	57.2 ± 30.5	46.8 ± 28.2	0.049 *
	JOA MCID success ratio (%)	58.2%	40.3%	0.038 *
Complications	Surgical site infection	1.5%	6.0%	0.091
	Dural tear	10.4%	3.0%	0.13
	CSF leakage	3.0%	1.5%	0.13
	Graft-related complications	10.4%	0%	0.001 *
	Upper airway obstruction	4.5%	0%	0.24
	Dysphagia	9.0%	0%	0.025 *
	C5 palsy	6.0%	1.5%	0.18
	Reoperation	9.0%	4.5%	0.25
VAS (2 y)	pain or stiffness in the neck or shoulder	35.9 ± 30.2	31.6 ± 28.1	0.43
	tightness in the chest	6.8 ± 18.0	7.9 ± 15.2	0.72
	pain or numbness in the arms or hands	41.4 ± 33.2	36.4 ± 27.3	0.39
	pain or numbness from chest to toe	31.0 ± 31.9	33.3 ± 32.5	0.71

ADF: anterior decompression with fusion; LAMP: laminoplasty; JOA: Japanese Orthopedic Association; RR: recovery rate, minimum clinically important difference (MCID); VAS: visual analogue scale; *: *p* < 0.05.

## Data Availability

The datasets generated during and/or analyzed during the current study are available from the corresponding author on reasonable request.

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
