# Peer review of "Prospective Investigation of Surgical Outcomes after Anterior Decompression with Fusion and Laminoplasty for the Cervical Ossification of the Posterior Longitudinal Ligament: A Propensity Score Matching Analysis"

_jcm, 2022, doi:10.3390/jcm11237012_

Round 1

Reviewer 1 Report

Good paper. The authors compared surgical outcomes of ACF with LAMP using a propensity score matching analysis.

The reviewer found this article to be interesting and totally agreed with their conclusion. However, the present version of the manuscript raised some points to be clarified and revised before justified for publication. The reviewer’s suggestion includes the followings.

Specific points

1.     What was the definition of graft complications?

2.     Did the authors investigate the local alignment at the ossification level? In addition to cervical alignment from C2 to C7, the local alignment may affect the results.

3.     In Table2, reoperation rate seemed to have significant p value.  If so, the authors should add the asterisk. What kind of reoperation was performed? What was the cause of reoperation?

4.     In propensity score matching analysis (Table 5), did the authors adjust the occupying ratio. Current evidence demonstrated anterior approach in OPLL is preferred for patients K-line (−) or patients with K-line (+) with canal occupying ratio >60%. Can the authors propose a more specific criteria including occupying ratio to make decisions in patients with cervical OPLL?

5.     There are some minor issues.

The writing of p is incomprehensive (p or p). Please check the manuscript thoroughly again.

The name of institution (line 82); ligamen is correct?

Author Response

Specific points

  1. What was the definition of graft complications?

Response: Thank you for this comment. Graft complication meant graft-related complication which included graft subsidence (≥2mm), graft dislodgement (≥2mm) as well as hardware failure such as plate dislodgement and screw back-out. We have described this definition in the Materials and Methods section. In addition, we have rephrased the term ‘graft complication’ with ‘graft-related complications’ to avoid misunderstanding.

  1. Did the authors investigate the local alignment at the ossification level? In addition to cervical alignment from C2 to C7, the local alignment may affect the results.

Response: We thank you for this valuable comment. Unfortunately, we did not have data of the local alignment at the ossification level. The reason of this is partly because number of ossification levels vary among the patients and also because multiple ossification lesions sometimes exist at the skipped levels (e.g. C2-3 and C6-7) in the same patient. Meanwhile, we agree that the local alignment is important and can affect the results. We have added this information to the limitation of this study.

  1. In Table2, reoperation rate seemed to have significant p value.  If so, the authors should add the asterisk. What kind of reoperation was performed? What was the cause of reoperation?

   Response: We apologize that we missed marking this. We have added the asterisk on it. Furthermore, we have added the causes and reoperation methods to the text as below. This is a very important point. We appreciate the reviewer for this suggestion.

There were 8 patients who received reoperation in the ADF group. Additional decompression was required for C5 palsy in 1 patient and for adjacent segmental disease in 1 patient. Hematoma removal was performed in 1 patient. Five patients required revision surgeries due to graft-related complications: anterior revision was performed for 2 patients and anterior revision followed by posterior fusion was required for 3 patients.

     There were 4 patients in the LAMP group who received reoperation. One patient with CSF leak required dural closure and 3 patient required debridement due to surgical site infection.

  1. In propensity score matching analysis (Table 5), did the authors adjust the occupying ratio. Current evidence demonstrated anterior approach in OPLL is preferred for patients K-line (−) or patients with K-line (+) with canal occupying ratio >60%. Can the authors propose a more specific criteria including occupying ratio to make decisions in patients with cervical OPLL?

   Response: We appreciate for this important comment. The reviewer is right. Both K-line and occupying ratio can affect the results. However, it was technically difficult and the model became unstable if we put many covariates which closely correlated with each other. Therefore, we chose K-line which covered both the size of OPLL and alignment as a covariant for propensity score calculating. Then, not only K-line but the occupying ratio became non-significant between ADF and LAMP. We have included this issue in the Discussion as a limitation of this study. Honestly, we did want to show some criteria for the decision making regarding the occupying ratio as the reviewer pointed out. However, we could not find significant tendency in the relationship between occupying ratio and surgical results in the presented cases. Thank you so much for this comment.

  1. There are some minor issues.

The writing of p is incomprehensive (p or p). Please check the manuscript thoroughly again.

The name of institution (line 82); ligamen is correct?

Response: Thank you so much for this comment. We have corrected the above issues.

Reviewer 2 Report

First of all, I would like to congratulate the authors. I think this is excellent work. The approach is correct, as well as the conclusions, supported by a proper bibliography. It is noteworthy how the authors are fully aware of the limitations of the work, which they have exposed in a clear and concise manner. 

However, I would like the authors to explain the justification for the 1:1 matching more clearly. 

Author Response

Specific points

First of all, I would like to congratulate the authors. I think this is excellent work. The approach is correct, as well as the conclusions, supported by a proper bibliography. It is noteworthy how the authors are fully aware of the limitations of the work, which they have exposed in a clear and concise manner. 

However, I would like the authors to explain the justification for the 1:1 matching more clearly. 

Response: We appreciate the reviewer for suggesting this important point. In the analysis using propensity scores, classical 1:1 matching is simple and easy for statistical comparison. Classical 1:1 matching without replacement may not be the most suitable approach when the number of individuals enrolled in a study is small. However, our study included relatively large sample size. It is reported that matching without replacement produced estimates with lower bias and better nominal coverage than matching with replacement when 1:1 matching was considered with enough amount of samples (BMC Med Res Methodol  2021 Nov 22;21(1):256.). Thus, we chose classical 1:1 matching using caliper (<0.2) without replacement. As a result, we could make a favorable matching between the ADF and LAMP group. We have added this explanation to the discussion section. We thank you for this valuable comment.
